# Pathophysiology and Treatment of Chronic Thromboembolic Pulmonary Hypertension

**DOI:** 10.3390/ijms24043979

**Published:** 2023-02-16

**Authors:** Naoyuki Otani, Ryo Watanabe, Takashi Tomoe, Shigeru Toyoda, Takanori Yasu, Takaaki Nakamoto

**Affiliations:** 1Department of Cardiology, Dokkyo Medical University Nikkyo Medical Center, Nikko 321-1298, Japan; 2Department of Cardiovascular Medicine, Dokkyo Medical University, Mibu 321-0293, Japan

**Keywords:** balloon pulmonary angioplasty, pulmonary endarterectomy, chronic thromboembolic pulmonary hypertension

## Abstract

Chronic thromboembolic pulmonary hypertension (CTEPH) is a condition in which an organic thrombus remains in the pulmonary artery (PA) even after receiving anticoagulation therapy for more than 3 months and is complicated by pulmonary hypertension (PH), leading to right-sided heart failure and death. CTEPH is a progressive pulmonary vascular disease with a poor prognosis if left untreated. The standard treatment for CTEPH is pulmonary endarterectomy (PEA), which is usually performed only in specialized centers. In recent years, balloon pulmonary angioplasty (BPA) and drug therapy for CTEPH have also shown good results. This review discusses the complex pathogenesis of CTEPH and presents the standard of care, PEA, as well as a new device called BPA, which is showing remarkable progress in efficacy and safety. Additionally, several drugs are now demonstrating established evidence of efficacy in treating CTEPH.

## 1. Introduction

Chronic thromboembolic pulmonary hypertension (CTEPH) is a condition that occurs in patients with pulmonary hypertension (PH), where an organic thrombus remains in the pulmonary artery (PA) even after being treated with anticoagulation therapy for more than 3 months; under the current clinical classification of PH, patients with a mean pulmonary arterial pressure higher than 20 mmHg CTEPH would be classified as group 4 PH [1]. CTEPH is a rare disease, with an estimated 2500 new cases occurring annually in the United States [2]. CTEPH is a progressive pulmonary vascular disease with a poor prognosis if left untreated. The number of patients diagnosed with CTEPH is increasing. Registry data indicates that the incidence and prevalence of CTEPH are 2–6 and 26–38 cases/million adults, respectively [3,4,5]. A potentially different approach has arisen in group 4 PH, where the concept of chronic thromboembolic disease without PH (CTED) has gained traction. In addition, the term chronic thrombo-embolic pulmonary disease (CTEPD) has been introduced for all patients whose symptoms can be attributed to post-thrombo-embolic fibrotic obstructions within the PA, with or without PH [6].

CTEPH is caused by stenosis or occlusion of the PA by an organic thrombus. It is estimated that 4% of acute pulmonary embolism cases become chronic and develop CTEPH within 2 years [7]. Therefore, CTEPH is thought to occur as a complication of acute pulmonary embolism following venous thromboembolism. It begins with persistent occlusion of the large- and medium-sized pulmonary arteries by an organic thrombus and is thought to be due to abnormalities of the fibrinolytic system and hematologic or autoimmune disease that prevented the thrombus from resolving [8]. However, it is becoming increasingly clear that microvasculopathy also contributes to hemodynamic compromise, dysfunction, and disease progression in CTEPH [8,9]. The molecular processes underlying microvasculopathy are not fully understood, and further research is needed. The standard treatment for CTEPH is pulmonary endarterectomy (PEA), which is usually performed only in specialized centers [10,11]. In recent years, balloon pulmonary angioplasty (BPA) and drug therapy for CTEPH have also shown good results. The CTEPH treatment algorithm includes a multimodal approach that combines PEA, BPA, and drug therapies to target mixed anatomical lesions such as proximal, distal, and microvasculopathy, respectively [12]. The pathogenesis of CTEPH is complex and involves multiple factors. This article reviews recent advances in understanding the pathophysiology of the CTEPH complex.

## 2. Pathology

In most cases of acute pulmonary thromboembolism, the thrombus in the PA dissolves; however, in some patients, it remains [13]. CTEPH is considered a late complication of acute pulmonary thromboembolism [7]. The International CTEPH Registry reported that 75% of patients with CTEPH had a history of acute pulmonary thromboembolism [14]. A characteristic feature of patients with CTEPH is obstructive fibrotic thromboembolic material in major pulmonary vessels [15]. It is noteworthy that the thromboembolic materials in acute pulmonary thromboembolism and CTEPH are distinct. Acute pulmonary thromboembolism is characterized by a red-colored, fresh thrombus with a fibrin network consisting mainly of red blood cells and platelets. Chronic thrombi seen in CTEPH are yellow in color, have a porous thrombus recanalization image composed of fibrous components such as collagen and elastin, inflammatory cells, and fibroblasts, and may rarely be accompanied by calcification [16]. The presence of multiple exacerbating factors, such as inflammatory factors and angiogenic disorders, contributes to the progression of an organic fibrotic thrombus [8]. CTEPH is characterized by progressive pulmonary vascular remodeling that develops in both occluded and non-occluded small PAs. In its most severe form, CTEPH can also present with focal capillary hemangiomatous and post-capillary venous remodeling as a consequence of bronchial-to-pulmonary shunting [10].

### 2.1. Inflammation and Infection

Inflammatory factors are also involved in the pathogenesis of CTEPH. Quarck et al. found that C-reactive protein (CRP) levels were higher in patients with CTEPH than in healthy controls and that CRP levels were significantly reduced after PEA. In this study, elevated CRP levels were not specific to CTEPH, and CRP levels were elevated in patients with PH [17]. Quarck et al. found that in addition to CRP, interleukin (IL)-10, monocyte chemoattractant protein (MCP-1), macrophage inflammatory protein (MIP)-1α, and matrix metalloproteinase (MMP)-9 were significantly elevated in patients with CTEPH, indicating a relationship between CTEPH and inflammation. Surgical specimens from patients undergoing PEA contained numerous macrophages, lymphocytes, and neutrophils, and correlations were found between CRP and neutrophil accumulation and between MMP-9 and macrophage accumulation [18]. Zabini et al. prospectively analyzed the serum of eight patients with CTEPH and reported significantly more elevated levels of IL-6, IL-8, interferon-γ-induced protein (IP)-10, monokine induced by interferon-γ (MIG), and MIP-1α, compared to those in age- and sex-matched healthy controls. Furthermore, in patients with CTEPH, IP-10 levels were negatively correlated with cardiac output, a 6 min walking distance, and carbon monoxide diffusing capacity, whereas IL-6 levels were positively correlated with pulmonary vascular resistance, right atrial pressure, and N-terminal pro-brain natriuretic peptide (NT-proBNP) [19]. Langer et al. measured the TNF-α levels in 14 patients with CTEPH before and after PEA. They reported that patients with CTEPH had higher TNF-α levels and that TNF-α levels rapidly normalized within 24 h after PEA [20].

Bonderman et al. proposed that staphylococcal infection promotes fibrovascular remodeling after thrombosis, impairs thrombolysis, and contributes to CTEPH development. All control samples (n = 28) were negative, but a staphylococcus-specific 420-bp product was present in 7 of the 26 patients with CTEPH clots, and in one of the patients with CTEPH with an existing suspicion of chronic infection (ventriculoatrial (VA) shunts for the treatment of hydrocephalus, or pacemaker carriers), Staphylococcus DNA was detected in six of seven clots collected during PEA [21]. In addition, Bonderman et al. conducted a case-control study comparing 109 patients with CTEPH with 187 patients with acute pulmonary thromboembolism and found that VA-shunts for the treatment of hydrocephalus (odds ratio 13, 95% confidence interval; 2.5 to 129) and splenectomy (odds ratio 13, 95% confidence interval; 2.7 to 127) were associated with an increased risk of CTEPH [22]. Moreover, Bonderman et al. performed a controlled retrospective cohort study using logistic regression modeling in patients with CTEPH (n = 433) and non-thromboembolic PH (n = 254) to investigate the prognostic relevance of predicted risk factors; VA shunt or infected pacemaker was a significant risk factor for developing CTEPH (odds ratio 76.40, 95% confidence interval; 7.67–10,351) [23]. In this study, splenectomy was also considered a risk factor for developing CTEPH (odds ratio 17.87, 95% confidence interval: 1.56 to 2438). Although the pathophysiological mechanisms linking splenectomies and CTEPH are not well understood, the fact that platelet-activating conditions in splenectomy are risk factors for CTEPH suggests a role for platelets in its genesis [8]. Splenectomies can affect the expression of various inflammatory cytokines, such as transforming growth factor-beta 1 (TGF-β 1), IL-2, and IL-10 [24,25], suggesting an effect on CTEPH.

### 2.2. Coagulation System Abnormality

Abnormalities in the coagulation system resulting in an over-coagulated state may be a factor in chronic thrombosis. Classic genetic thrombogenic risk factors, such as protein C, protein S, and antithrombin deficiency, were not significantly different between the CTEPH and control groups [26]. On the other hand, lupus anticoagulant/antiphospholipid antibodies showed more association in patients with CTEPH than in those with non-thromboembolic PH (odds ratio 4.20, 95% confidence interval: 1.56–12.21) [23]. High plasma levels of factor VIII are a risk factor for venous thromboembolism [27]. In a study comparing plasma factor VIII in 122 patients with CTEPH with 82 healthy subjects and 88 patients with non-thromboembolic pulmonary arterial hypertension (PAH), the CTEPH group had higher levels than the healthy subjects (233 ± 83 IU/dl versus 123 ± 40 IU/dl, *p* < 0.0001) and non-thromboembolic PAH (158 ± 61 IU/dl, *p* < 0.0001) [28]. This elevation in plasma factor VIII was suggested to be a possible prothrombotic factor in patients with CTEPH.

### 2.3. Fibrinolytic Abnormality

Abnormalities in fibrinogen may not only promote thrombus formation but also slow its degradation and facilitate its conversion from thrombolysis to thrombus formation. Genetic polymorphisms in fibrinogen have been reported to be involved in the insolubilization of blood clots. Significant differences were observed between CTEPH and controls in the frequency of the Thr312Ala (rs6050) polymorphism (*p* = 0.03) and alleles (*p* = 0.01) in the fibrinogen alpha chain [29]. As the fibrin structure degenerates, it acquires resistance to plasmin fibrinolysis, which may contribute to chronic thrombosis [30]. In the study by Olman et al. [31], neither an increase in type 1 plasminogen activator inhibitor nor a blunted response of tissue-type plasminogen activator was observed. Yaoita N et al. studied the role of thrombin-activatable fibrinolysis inhibitor (TAFI), which inhibits fibrinolysis, in the pathogenesis of CTEPH in 27 patients with CTEPH, 22 with PAH and 19 controls without PH. Plasma TAFI levels were significantly higher in patients with PAH (19.4 ± 4.2 μg/mL versus 16.1 ± 4.5 μg/mL, *p* < 0.05) or controls (16.3 ± 3.3 μg/mL, *p* < 0.05) than those in patients with CTEPH [32].

### 2.4. Abnormal Platelet Function

Patients with CTEPH have decreased platelet counts, increased mean platelet volume, increased spontaneous platelet aggregation, and decreased induced platelet aggregation [33]. This suggests that platelet activation, consumption, or destruction occurs chronically in patients with CTEPH. In addition, the GTP-binding GTPase RhoA, which is involved in the surface expression of P-selectin, PAC-1 binding, and platelet aggregation, is activated on platelets in the CTEPH group compared to that in the non-CTEPH group [34]. The platelet factor 4 level is an indicator of platelet activation. Tissues removed by PEA from patients with CTEPH had higher platelet factor 4 levels than those in healthy lung tissue. Elevated levels of platelet factor 4 may be associated with reduced angiogenesis [35]. Platelet Endothelial Cell Adhesion Molecule (PECAM)-1 is specifically distributed on platelets, leukocytes, and endothelial cells and is associated with thrombosis and inflammation [36]. Comparing patients with deep-vein thrombus with lysed thrombus and patients with deep-vein thrombus with delayed lysis, patients with delayed thrombus lysis had significantly higher plasma PECAM-1 level than those with lysed thrombus [37]. Thus, PECAM-1 levels in humans after acute deep-vein thrombosis may correlate with thrombolysis.

### 2.5. Cancer

Thrombosis is a common complication in cancer patients, with an estimated 20% of cancer patients experiencing venous thromboembolism [38,39]. Findings from a European database including 687 patients with CTEPH (n = 433), and non-thromboembolic PH (n = 254) support an association between a history of malignancy and CTEPH (odds ratio 3.76, 95% confidence interval 1.47 to 10.43, *p* = 0.005) [23].

### 2.6. Blood Groups

Patients with CTEPH commonly have non-O blood types (A, B, or AB). A European database study suggested that non-O blood type was a significant predictor of CTEPH diagnosis (odds ratio 2.09, 95% confidence interval; 1.12–3.94; *p* = 0.019) [23]. Wu et al. performed a meta-analysis and reported that non-O blood types are at higher risk for deep vein thrombosis than O blood types (odds ratio 1.79, 95% confidence interval 1.56–2.05) [40].

### 2.7. Myofibroblasts

Myofibroblasts are present within the organizing thrombus that is removed by PEA [41]. Myofibroblasts may be due to fibroblast differentiation in pulmonary vessels, fibrocytes accumulating from peripheral blood into the lesion, or epithelial-mesenchymal transition [42]. Myofibroblasts are thought to deposit extracellular matrix and are involved in chronic fibrosis [43].

### 2.8. Impaired Angiogenesis

It has been reported that there are fewer neovessels in organizing thrombi obtained by PEA. This lack of neovascularization suggests that it is an important mechanism of occlusive vascular remodeling after deep vein thrombosis [44]. Neovascular suppressors such as PF4, collagen type I, and IP-10 are increased in specimens removed from patients with CTEPH after PEA, which may result in inadequate thrombus recanalization [35].

### 2.9. Small-Vessel Disease

Occlusion of the central PA by a thrombus is the first trigger for the development of CTEPH [45]. This is followed by occlusion of the peripheral pulmonary arteries. Occlusion of many pulmonary arteries increases blood flow to non-obstructed vessels and shear stress on the vessel wall. This results in pulmonary vascular remodeling and PH [46]. When a central PA is completely occluded by a thrombus over a long period, the tunica media may atrophy on its peripheral side, and the muscular arteries may change to vein-like structures [47]. In healthy individuals, body vessels, such as bronchial arteries or vas deferens, undergo hemodynamic bypass with peripheral pulmonary arteries and pulmonary veins, respectively. At the periphery of the occluded PA, the pressure gradient with the body vasculature increases through this bypass, increasing the peripheral PA shear stress. In addition, blood flow increases from the body vessels directly into the pulmonary venous system, and the venous system responds to pulmonary capillary hemangioma-like changes and pulmonary vein occlusive disease-like vascular lesions [10].

## 3. Diagnosis of CTEPH

The initial investigations in suspected CTEPH should include an echocardiogram and a ventilation-perfusion (V/Q) scan. Enhanced computed tomography (CT) has been considered useful in diagnosing CTEPH but advances in treating peripheral forms of CTEPH have increased the importance of V/Q scan. The diagnosis was made if there was evidence of residual organizing thrombus in the PA and PH after at least 3 months of anticoagulation to exclude the possibility of acute pulmonary thromboembolism. Right heart catheterization is essential to confirm that the hemodynamics of the pulmonary circulation meet the diagnostic criteria (mean PA pressure > 20 mmHg and PA wedge pressure ≤ 15 mmHg).

CTEPH can mimic acute pulmonary embolism at presentation [48]. Thus, CTEPH can be misdiagnosed as acute pulmonary embolism, which is one of the reasons for delayed diagnosis. A CTEPH risk score has been developed to help stratify the risk of CTEPH and guide the need for further investigations (Table 1) [4]. A strategy using this score accurately excluded CTEPH without the need for echocardiography in the majority of pulmonary embolism patients, resulting in a substantially shorter diagnostic delay than in current practice.

### 3.1. Blood Tests and Immunology

Elevated levels of brain natriuretic peptide (BNP), NT-proBNP, and uric acid are associated with right ventricular overload and are predictors of a poor prognosis. Abnormal liver function may be caused by congestion, primary liver disease, or treatment effects. To search for coagulation abnormalities, thrombogenic markers such as D-dimer, protein C, and protein S, and autoantibodies involved in thrombus formation, such as lupus anticoagulant and anticardiolipin antibodies, have been measured [49].

### 3.2. Pulmonary Function Tests

Lung function tests should include the total lung capacity and diffusing capacity of the lung for carbon monoxide (D_LCO_). In most patients with PAH, there is a mild restraint disorder. A decreased D_LCO_ may also be observed in patients with CTEPH [50].

### 3.3. Echocardiography

Echocardiography remains the most important noninvasive screening tool, and right heart catheterization (RHC) is mandatory to establish a diagnosis [51,52].

### 3.4. (V/Q) Scanning

Pulmonary V/Q scanning has become popular among clinicians as a noninvasive diagnostic method for identifying patients with pulmonary vascular disease. Owing to the sensitivity of the V/Q scan, the presence of CTEPH can be ruled out if the V/Q scan is normal [53]. In addition, CT pulmonary angiography can complement the information obtained from pulmonary V/Q scans and provide additional data on anatomic localization and surgical potential.

### 3.5. Chest Computed Tomography (CT)

CT is a useful method for ruling out CTEPH. In patients presenting with a clinical picture of acute pulmonary embolism, chest CT may be helpful in detecting signs of hitherto undetected CTEPH [48]. It is also considered useful in detecting subsegmental or segmental artery thrombosis, which is necessary to determine the indication for surgery in patients undergoing PEA [54]. Digital subtraction angiography is still used to assess treatment options when CT pulmonary angiography is inconclusive. Selective segmental angiography, cone-beam CT, and area-detector CT allow for more accurate visualization of subsegmental vasculature and are useful for procedural guidance for BPA [11]. Measurement of the main PA diameter and PA-to-aorta ratio using CT may play an important role in diagnosing PH [53].

### 3.6. Right Heart Catheterization (RHC)

RHC is an essential test for the definitive diagnosis of PH. In central-type CTEPH, PA pressure may differ locally due to central organizing thrombus; therefore, it should be measured in the left and right main pulmonary arteries and PA trunks. In addition, measurement of PA wedge pressure may be difficult because of the adhesion of the organic thrombus, and the waveform should be thoroughly checked before measurement.

### 3.7. Pulmonary Digital Subtraction Angiography (PDSA)

Traditionally, RHC and PDSA have been standard methods used to diagnose CTEPH. PDSA has been considered the gold standard for characterizing the vascular morphology of CTEPH; however, noninvasive treatments are advancing. CT pulmonary angiography is widely used to assess operability. Compared with PDSA, the sensitivity and specificity of CTEPH findings were 97.0% and 97.1% (*p* < 0.001) at the main/foveal level, respectively, and 85.8% and 94.6% (*p* < 0.001) at the branch level [55]. With advances in BPA, more attention is being paid to distal vascular evaluations. The use of cone-beam CT and electrocardiogram-gated area-detector CT has enabled the visualization of distal lesions in CTEPH [56]. These new imaging techniques may be useful for evaluating target lesions prior to BPA exposure.

## 4. Surgical Treatment: Pulmonary Endarterectomy (PEA)

PEA is the standard treatment for CTEPH, which involves surgical removal of the clot and is usually performed only in specialized centers [10]. Therefore, after confirming the diagnosis of CTEPH, the indication for surgery with PEA is determined first. The indication for PEA is determined by morphological characteristics defined by pulmonary angiography, CT, or magnetic resonance imaging (MRI) and the results of hemodynamic evaluation by right heart catheterization [57]. The selection criteria for PEAs are listed in Table 2 [58]. Surgery is indicated for patients with central-type CTEPH, which affects the central pulmonary arteries, including the main pulmonary, interlobar, and segmental arteries, causing mural thrombus and intimal hyperplasia. However, effective surgical treatment may not be possible for patients with peripheral CTEPH. In 2012, the University of California, San Diego classified PEA indication criteria into four types of PA occlusion morphology based on the excised thrombus intima (Table 3) [59]. Types I and II are central and are good indications for PEA. Type III is the peripheral type, and the surgical technique is more difficult. Therefore, only a limited number of cases are indicated for PEA. Type IV is a lesion of the small arteries that is not an indication of surgery. Patients with distal thromboembolism type three or four had higher perioperative mortality than those with type one or two (3.9% and 4.7%, respectively) [58]. Subsequently, the distal limits of endarterectomy were redefined, and a new level classification was reported by the University of California, San Diego, which better reflects revascularization (Table 4) [60,61].

PEA is a major procedure performed through median sternotomy under very hypothermic (16–18 °C) intermittent circulatory arrest. Incise the left and right main pulmonary arteries, find an appropriate dissection surface between the internal elastic plate and tunica media, proceed to the regional artery, and remove the organizing thrombus. If the organizing thrombus strongly adheres to the PA wall, it should be removed along with the PA intima. Given that the key to performing surgery is a dissection of the intima, it is necessary to remove the organizing thrombus along with the intima. Common immediate post-operative complications of PEA are reperfusion lung injury and intratracheal hemorrhage. Reperfusion lung injury typically occurs within 48 h of PEA. Indeed, it was reported to occur in 9.6% of patients in a large European/Canadian registry [10]. In some cases, veno-venous ECMO may be necessary for temporary support in patients with the most severe reperfusion injury [62]. However, one-third to one-half of patients can still be salvaged and discharged [63]. Veno-arterial ECMO is necessary if there is hemodynamic instability, particularly in cases of persistent PH [64]. Heparin should be initiated once the risk of bleeding decreases. Residual postoperative PH may require management of right heart failure with pulmonary vasodilators or catecholamines. If the heart failure is severe, an artificial heart-lung or aortic balloon pump may be used. Some patients with severe right heart failure or reduced cardiac output have difficulty weaning from cardiopulmonary bypass, and ECMO (extracorporeal membrane oxygenation) has been used to support patient hemodynamics. ECMO is required in about 5% of patients in large surgical series [4]. The period of support has ranged from 49 to 359 h, with a mean duration of 119 h and survival of up to 57% [65]. PEA outcomes are favorable due to improved management of cardiac and pulmonary complications and the well-established use of ECMO. In 2014 and 2015, 394 PEAs were performed in Germany with an in-hospital, periprocedural, (0%) and in-hospital (2.5%) mortality rates that were very low [66]. Mortality correlated with mean PA pressure at diagnosis. In 2001, Riedel M et al. reported that the 5-year survival rate for patients with mean PA pressure greater than 50 mmHg is approximately 10%, and the prognosis for patients with mean PA pressure less than 30 mmHg was much better [67].

Thistlethwaite et al. reported the surgical outcome of 1100 PEA for CTEPH and reported an overall mortality rate of 4.7% [58]. The International CTEPH Registry reports a perioperative-related mortality rate of 3.4% at centers that perform more than 50 PEAs per year, with remote outcomes after surgery of 93% survival rate at 1 year, 91% at 2 years, and 89% at 3 years [68]. In a 2009–2016 report in Japan, the 5-year survival rate was 89.9%, which is good [69]. On the other hand, 31% of patients had residual PH after PEA [70], and there was no effective treatment for these patients with residual PH.

CTEPH is inoperable in at least 20–40% of patients because of distal disease or comorbidities [71]. If none of the treatments are effective, lung transplantation can be considered as a salvage option only in highly selected patients and in highly specialized centers [72]. In the case of CTEPH the lung-transplant should always be bilateral to avoid the entire flow of the right ventricle from being directed into a single vascular bed with the subsequent high-risk of reperfusion edema. The 2019 International Society for Heart and Lung Transplantation (ISHLT) registries reported that only 1.5% of lung transplants performed worldwide are for non-idiopathic PH, so we can assume that the percentage of transplants performed for CTEPH is even lower. Lung transplant for CTEPH can be considered as a salvage option, only in highly selected cases as an alternative to failed PEA or as curative treatment in very peripheral thromboembolic disease not amenable to other drug or surgical treatment [73]. Some patients with CTEPH may have mixed anatomic lesions, with operable lesions in one lung and inoperable lesions in the other lung. For these patients, the combination of PEA and BPA before or simultaneously with PEA may reduce the risk of surgery and improve the outcome [74].

In recent years, surgical outcomes for PEA have improved, and surgical treatment is now being considered even for patients with significant chronic vascular obstruction but with near-normal pulmonary hemodynamics at rest. Taboada et al. defined this group as “chronic thromboembolisms” (CTED). They reported that CTEDs significantly improved symptoms and quality of life with PEA only in centers with carefully selected surgical indications and extensive surgical experience [75]. Selected symptomatic patients with CTEPD without PH can be successfully treated by PEA, with clinical and hemodynamic improvements at rest and during exercise [75,76].

## 5. Interventional Treatment: Balloon Pulmonary Angioplasty (BPA)

BPA is a percutaneous catheter intervention that mechanically dilates stenosis and obstructs the PA with a balloon catheter. Invasive treatment of congenital PA stenosis by catheterization of the PA began in the late 20th century. Subsequently, catheterization was initiated for the stenotic and occluded lesions due to CTEPH. In 2001, Feinstein et al. reported 18 patients with inoperable CTEPH treated with BPA. BPA improved hemodynamic measures, including pulmonary arterial pressure, but the results were inferior to those of PEA. In addition, a high rate of reperfusion pulmonary edema, which is occasionally fatal, occurs after BPA [77]. The treatment outcomes of BPA do not exceed the efficacy and safety of PEA in skilled centers. Therefore, BPA has not been emphasized as a treatment option for CTEPH in the U.K. and U.S. On the other hand, in Japan, the percentage of patients with peripheral CTEPH is higher than in Europe and the United States, and many of these cases are not amenable to surgical treatment. In 2012, a few centers in Japan reported that a refined version of BPA demonstrated a significant improvement in efficacy and safety compared to the initial report [78,79]. Subsequently, BPA studies have been reported by many centers worldwide, and in 2017, a multicenter registry study from Japan showed similar efficacy and safety to those of single-center studies [80]. Although BPA is an invasive procedure with accompanying risks, it is less invasive than surgery. BPA techniques have been refined since their introduction in 2001, and BPA is now widely used worldwide.

PEA is an established curative treatment for CTEPH [11], but this procedure requires advanced surgical skills and surgical experience. Therefore, approximately half of the patients with CTEPH remain inoperable. These cases are known to have poor prognoses. Therefore, the indication for BPA is basically “cases in which PEA is not indicated and drug treatment is ineffective,” and the conditions shown in Table 5 should be met to perform BPA.

When BPA is performed, preoperative management is important to avoid intraoperative and postoperative complications. Whenever possible, efforts should be made to improve and stabilize hemodynamic and systemic status prior to BPA. Patients receiving oral vasodilators preoperatively continue receiving them, and for severe cases with low cardiac output with a cardiac coefficient of less than 2.2 L/min/m^2^. Inotropic drugs such as dobutamine should be administered preoperatively.

The therapeutic goal of BPA, similar to PEA, is thought to be the elimination of PH, prevention of the development of right heart failure, and improvement of life expectancy. The optimal goal of BPA might be to achieve a mean pulmonary arterial pressure of less than 25 mmHg (<20 mmHg if possible) and to wean the patient off home oxygen therapy by improving oxygenation. Therefore, it is essential to perform complete revascularization of all PA lesions as close as possible. A clinical trial in the UK found that clinical outcomes correlated with hemodynamic change (pulmonary vascular resistance [ΔPVR] vs. Cambridge Pulmonary Hypertension Outcome Review [CAMPHOR] symptom score: *p* < 0.001, Pearson’s r = 0.48), which is related to the number of segments treated and lesion severity. Responders to BPA had more severe disease at baseline, and 37.5% of non-responders were post-PEA. Treatment of completely occluded vessels has a greater hemodynamic impact [81]. In addition, a clinical trial from Austria found that the number of successfully treated vascular lesions predicts treatment response to BPA and that the number of successfully recanalized occlusions (particularly chronic total occlusions) appears to have the strongest impact on changes in mean PA pressure [82]. A recent study in Japan compared the safety and efficacy of BPA and riociguat in patients with inoperable CTEPH. Compared with riociguat, BPA was associated with a greater improvement in mean pulmonary arterial pressure in patients with inoperable CTEPH at 12 months [83]. A systematic review and meta-analysis suggest that BPA may provide greater functional and hemodynamic improvement than pulmonary vasodilators [84]. Recently, BPA has become an established treatment for selected patients with inoperable CTEPH or persistent/recurrent PH after PEA, but evidence is not yet sufficient.

Although BPA is effective, it is associated with serious complications, which may be fatal. The most common complication associated with BPA is procedure-related lung injury (incidence: 53–60%) [79]. From a technical standpoint, BPA does not differ significantly from balloon angioplasties performed on other vessels. However, it requires special experience owing to the complex anatomy of the pulmonary field, the need for instruments to pass through the enlarged right ventricle, and the possibility of pulmonary vessels being easily damaged by guidewires or balloon catheters [85]; therefore, this procedure should be performed in high-volume CTEPH centers. As the rates of interventional complications can be reduced by medical pre-treatment, patients with a PVR > 4 WU should be treated before BPA [86].

BPA, currently recommended for patients who are not candidates for PEA, is expected to become an increasingly common treatment. However, BPA does not replace PEA. As interventional techniques improve, BPA may become the procedure of choice for patients at high perioperative risk, even when surgery is indicated. There are also hopes for hybrid procedures consisting of cardiac surgical resection of accessible lesions and interventional treatment of distant lesions. Although data regarding the safety and efficacy of BPA in patients with CTEPD are lacking, carefully selected symptomatic patients with CTEPD without PH and segmental/subsegmental lesions can be successfully treated by BPA, with clinical and hemodynamic improvements at rest and during exercise [11,75,76].

## 6. Drug Therapy 

PEA is a curative treatment; however, drug therapy is necessary for patients with inoperable disease or persistent or recurrent PH after PEA. In a survey of five PH centers in the U.K. from January 2001 to June 2006, one- and three-year survival rates from diagnosis were 82% and 70% for patients with nonsurgical disease and 88% and 76% for those treated surgically (*p* = 0.023). Drug treatment resulted in initial functional improvement, which was not sustained after 2 years [87].

Drug therapy for CTEPH includes anticoagulation therapy, oxygen therapy, and pulmonary vasodilators. Lifelong therapeutic anticoagulation is recommended for patients with CTEPH, as recurrent pulmonary thrombo-embolism accompanied by insufficient clot resolution are key pathophysiological features of this disease. Anticoagulation with warfarin is recommended by experts and is most widely used as basic therapy for patients with CTEPH. It should be administered for life, with a controlled PT-INR of 2.0–3.0 [88]. However, there are no randomized controlled trials (RCTs) in CTEPH with any of the approved anticoagulants. Recently, direct oral anticoagulants have more frequently been used as alternatives to warfarin; the usefulness of direct oral anticoagulants has not yet been reported. A retrospective case series from the UK and a multicenter prospective registry (EXPERT) showed comparable bleeding rates for warfarin and direct oral anticoagulants in CTEPH, but recurrent venous thrombo-embolism rates were higher in those receiving direct oral anticoagulants [89,90]. Oxygen therapy, including home oxygen therapy, should be administered for hypoxemia with symptoms such as shortness of breath on exertion.

The soluble guanylate cyclase stimulator riociguat is currently the only pharmacotherapy approved by the United States Food and Drug Administration (FDA). Indications for drug therapy include cases in which BPA and PEA are not indicated and cases of residual PH after BPA and PEA; however, it has recently been used as a bridge to BPA and PEA. The clinical and pathological similarities between CTEPH and PAH suggest that targeting PAH may be effective. Currently, the three targets of PAH treatment are the endothelin, nitric oxide, and prostacyclin pathways [91].

### 6.1. Prostacyclin Therapy

#### 6.1.1. Epoprostenol

Cabrol et al. retrospectively analyzed 27 patients with inoperable distal CTEPH who received long-term intravenous epoprostenol. Compared with baseline after 3 months of receiving epoprostenol, 11 of 23 patients showed improvement in NYHA Functional Class, a 6 min walking distance increased by 66 m (*p* < 0.0001), and hemodynamics also improved (mPAP, *p* < 0.001; cardiac index, *p* = 0.0003; total pulmonary resistance, *p* < 0.0001). The 1-, 2-, and 3-year survival rates were 73%, 59%, and 41%, respectively [92]. Abe et al. prospectively randomized patients with residual PH after PEA to inhaled nitric oxide (n = 6) or inhaled epoprostenol sodium (n = 7). Observed from before drug inhalation to the day after receiving, mean pulmonary arterial pressure and pulmonary vascular resistance decreased significantly over time in both groups (mean pulmonary arterial pressure: *p* < 0.0001, pulmonary vascular resistance: *p* = 0.003) [93]. In a study that observed whether prostacyclin therapy improves PH in patients with CTEPH prior to undergoing PAH, receiving intravenous prostacyclin reduced pulmonary vascular resistance by 28%, from 1510 ± 53 wood units to 1088 ± 58 wood units (*p* < 0.001), and plasma BNP levels significantly decreased from 547 ± 112 pg/mL to 188 ± 30 pg/mL (*p* < 0.01) [94].

#### 6.1.2. Iloprost

Krug et al. prospectively studied the acute hemodynamic effects of inhaled iloprost in 20 patients with CTEPH. After inhalation of iloprost, pulmonary vascular resistance decreased from 1057 ± 404.3 to 821.3 ± 294.3 wood units (*p* < 0.0001), mean pulmonary arterial pressure decreased from 50.55 ± 8.43 to 45.75 ± 8.09 mm Hg (*p* = 0.0002), cardiac output increased from 3.66 ± 1.05 to 4.05 ± 0.91 L/min (*p* < 0.0106) [95]. Kramm et al. evaluated the hemodynamic effects of inhaled iloprost before PEA and in the early postoperative period in 10 patients with CTEPH. However, iloprost inhalation did not alter the mean pulmonary arterial pressure, heart rate, or pulmonary vascular resistance [96]. There are no long-term results of iloprost inhalation in patients with CTEPH and few clinical studies with a large number of cases.

#### 6.1.3. Beraprost

Ono et al. reported 43 patients with peripheral vascular CTEPH: 20 receiving beraprost and 23 not receiving beraprost. Ten patients in the group receiving beraprost had improved NYHA functional class and significantly reduced total pulmonary resistance from 18 ± 6 to 15 ± 8 Wood units (*p* < 0.05). The 1-, 3-, and 5-year survival rates were 100%, 85%, and 76% for patients receiving beraprost, compared to 87%, 60%, and 46% for patients not receiving beraprost [97].

#### 6.1.4. Treprostinil

Patients with CTEPH classified as inoperable according to the WHO functional class III or IV or with persistent or recurrent PH after PEA were randomized to receive high-dose treprostinil subcutaneously (53 patients) or low-dose treprostinil subcutaneously (52 patients). The primary endpoint was the change from baseline in the 6 min walking distance at week 24. Mean 6 min walking distance improved by 44–98 m (95% CI 27–52 to 62–45) in the high-dose group and by 4–29 m (95% CI −13–34 to 21–92) in the low-dose group mean 6 min walking distance improved by 44–98 m (95% CI 27–52 to 62–45) in the high-dose group and by 4–29 m (95% CI −13–34 to 21–92) in the low-dose group (treatment effect 40–69 m, 95% CI 15.86 to 65.53, *p* = 0.0016). The most common treatment-related adverse events in both groups were injection site pain and other injection site reactions [98]. In an open-label, uncontrolled study, Skoro-Sajer et al. examined treprostinil treatment in 25 patients with inoperable CTEPH of WHO functional class III or IV. The control group was a historical group of 31 patients with inoperable CTEPH, matched for severity. Treprostinil-treated patients had improved 6 min walking distance (*p* = 0.01), WHO functional class (*p* = 0.001), B-type brain natriuretic peptide plasma levels (*p* = 0.02), cardiac output (*p* = 0.007), and pulmonary vascular resistance (*p* = 0.01) [99]. Treprostinil, a prostacyclin analog, has been approved for use in patients with CTEPH in the EU.

### 6.2. Endothelin Receptor Antagonists

#### 6.2.1. Bosentan

Chen et al. presented a meta-analysis of RCTs on bosentan treatment for CTEPH. Ten RCTs involving 1185 patients were enrolled. Bosentan improved cardiac index by 0.3 L/min/m^2^ and reduced pulmonary vascular resistance by 176.0 dyn-s/cm^5^. However, there were no statistical differences in other efficacy measures related to CTEPH. No significant differences were observed in mortality or adverse events between bosentan and placebo groups. However, bosentan increased the risk of liver function abnormalities. Therefore, the authors concluded that bosentan improves only certain hemodynamic parameters of CTEPH [100].

In a large, randomized clinical trial in patients with inoperable CTEPH or persistent/recurrent pulmonary hypertension after PEA (>6 months after PEA), the BENEFiT (Bosentan Effects in iNopErable Forms of chronIc Thromboembolic pulmonary hypertension) trial was conducted [101]. This was the first multicenter, randomized, placebo-controlled trial of only patients with CTEPH (inoperable CTEPH or PH lasting more than 6 months after PEA) and included 157 patients, 28% of whom had previously undergone PEA. Eighty patients were assigned to the placebo group and 77 to the bosentan group, with the primary endpoint being the percentage change in pulmonary vascular resistance and 6 min walking distance after 16 weeks relative to baseline. After 16 weeks of treatment, mean pulmonary vascular resistance decreased from baseline in the bosentan group and increased in the placebo group (therapeutic effect on pulmonary vascular resistance: −24.1%; 95% CI: −31.5~−16.0; *p* < 0.0001). However, there was no statistically significant improvement for bosentan compared to placebo, with a mean change in 6 min walking distance from baseline to week 16 of +2.9 m in the bosentan group and +0.8 m in the placebo group (Average treatment effect = +2.2 m (95% CI: −22.5 to 26.8; *p* = 0.5449)). The reason for the discrepancy between the effects of bosentan on these two endpoints, hemodynamic indices, and exercise tolerance is unclear. Because the BENEFiT trial did not meet its primary endpoint, bosentan was not approved for use in patients with CTEPH in the United Kingdom, the United States, and Japan.

#### 6.2.2. Macitentan

Macitentan, a dual ETRA/ETRB antagonist, was designed by modifying the structure of bosentan, and approved for PAH. The phase 2, double-blind, randomized, placebo-controlled MERIT-1 trial assessed the efficacy of macitentan at a dose of 10 mg in 80 patients with CTEPH adjudicated as inoperable. At 16 weeks, geometric mean PVR decreased to 73.0% of baseline in the macitentan group and to 87.2% in the placebo group. Macitentan significantly improved PVR in patients with inoperable CTEPH [102]. A phase 3 RCT is ongoing to evaluate efficacy and safety of macitentan 75 mg in inoperable or persistent/recurrent CTEPH (MACiTEPH). This study is registered with ClinicalTrials.gov. number NCT04271475.

### 6.3. Phosphodiesterase Type-5 Inhibitors

In an open-label uncontrolled clinical trial by Reichenberger et al., 104 patients with inoperable CTEPH received sildenafil 50 mg three times daily. Pulmonary vascular resistance at baseline was 863 ± 38 dyn.s.cm^−5^, and after 3 months of treatment, pulmonary vascular resistance was 759 ± 62 dyn.s.cm^−5^ (*p* = 0.0002 versus baseline), indicating significant hemodynamic improvement. The 6 min walking distance significantly increased from 310 ± 11 m at baseline to 361 ± 15 m at 3 months (*p* = 0.0001 versus baseline) and 366 ± 18 m at 12 months (*p* = 0.0005 versus baseline) [103]. In this double-blind, placebo-controlled pilot study of sildenafil reported by Suntharalingam et al., 19 patients with inoperable CTEPH or persistent PH after PEA were enrolled. The primary endpoint, change in 6 MWD at 12 weeks, was not significantly different between the sildenafil and placebo groups. The authors believe that the study was small and had insufficient power. There was a significant improvement in the NYHA functional class and pulmonary vascular resistance (−197 dyn.s.cm^−5^, *p* < 0.05) in the sildenafil group. After completion of the study, the patients were transferred to an open-label clinical study of sildenafil and re-evaluated 12 months later. Sildenafil administration significantly improved exercise capacity, pulmonary vascular resistance, and NT-proBNP levels compared with baseline values [104]. Although sildenafil has been used off-label, as its efficacy in inoperable CTEPH has not been proven by RCTs or registry data, oral combination therapy, including phosphodiesterase type-5 inhibitors and endothelin receptor antagonists, is common practice in patients with CTEPH with severe hemodynamic compromise [105].

### 6.4. Soluble Guanylate Cyclase Stimulator

#### Riociguat

Riociguat stimulates the NO-sGC-cGMP pathway, inhibits smooth muscle cell contraction, and promotes vasorelaxation. Owing to its mechanism of action, it is expected to be applied to microvascular disorders that cannot be approached by BPA or PEA [106].

Ghofrani et al. performed a 12-week, multicenter, open-label phase II study in 42 patients with World Health Organization (WHO) functional class II/III CTEPH. Patients with CTEPH treated with riociguat had a median 6 min walking distance increase of 55 m from baseline (*p* < 0.0001) [107]. In addition, Ghofrani et al. demonstrated the Chronic Thromboembolic Pulmonary Hypertension Soluble Guanylate Cyclase–Stimulator Trial 1 (CHEST-1) in a phase 3, multicenter, randomized, double-blind, placebo-controlled study. A total of 261 patients with inoperable chronic thromboembolic PH or persistent or recurrent PH after pulmonary endarterectomy were randomly assigned to receive a placebo or riociguat. The primary endpoint was the change from baseline to the end of week 16 at a 6 min walking distance. By week 16, the 6 min walking distance increased an average of 39 m in the riociguat group, and it decreased an average of 6 m in the placebo group (least-squares mean difference, 46 m; 95% confidence interval, 25 to 67; *p* < 0.001). Riociguat also showed significant improvements in secondary endpoints such as pulmonary vascular resistance, NT-proBNP, and NYHA functional class [108]. An open-label extension study (CHEST-2) was conducted with 243 patients who completed the CHEST-1 study, enrolling 237 patients. After 1 year, the improvement in 6 MWD and WHO functional class seen in the CHEST-1 trial was maintained [109].

Riociguat is the world’s first approved pharmacotherapy for CTEPH, which is currently approved for the treatment of CTEPH, with approval in the USA and the EU, as well as in many other countries, including Canada, Australia, and Japan. Currently, both riociguat and BPA are recommended for inoperable CTEPH [14,110,111]. The 2015 European Society of Cardiology/European Respiratory Society guidelines recommended riociguat for treating inoperable CTEPH [92]. The 2017 Japanese Circulation Society guidelines recommended riociguat for treating inoperable CTEPH [110].

### 6.5. Bridge to PEA

Receiving a treatment of PAH as a bridge therapy to PEA is recognized by reports of improved postoperative outcomes by decreasing pulmonary vascular resistance before surgery.

Nagaya et al. administered prostacyclin therapy to patients with severe CTEPH as a pretreatment for PEA. Intravenous prostacyclin reduced pulmonary vascular resistance and plasma BNP levels in patients with CTEPH. In addition, a clinical trial demonstrated a relatively low surgical mortality rate (8.3%) in patients with severe CTEPH receiving prostacyclin [94]. Bresser et al. also retrospectively evaluated the clinical and hemodynamic responses to continuous intravenous epoprostenol in nine patients with CTEPH who subsequently received PEA. Six patients treated for 2 to 26 months prior to PEA experienced clinical stability or improvement, associated with an average 28% reduction in pulmonary vascular resistance (median 33%, range 0–46%) [112]. Reesink et al. conducted a randomized, controlled, single-blind study. The patients were randomized to receive bosentan (n = 13) or no bosentan (n = 12) for 16 weeks. The primary endpoint was the change in total pulmonary resistance from baseline to 6 weeks, with a mean difference in total pulmonary resistance between the two groups of 299 WU (*p* = 0.004). The mean difference in the secondary endpoint of the 6 min walking distance was 33 m (*p* = 0.014), and the mean PA pressure was 11 mmHg (*p* = 0.005) [113]. Jensen et al. retrospectively analyzed the medical treatment of patients with CTEPH who were referred for PEA between 2005 and 2007. The control group (n = 244) was compared with the PHT group (n = 111), and the subgroups included monotherapy and combination therapy with bosentan, sildenafil, and epoprostenol. The use of targeted therapy for preoperative pulmonary vascular PH significantly increased from 19.9% in 2005 to 37% in 2007. However, it was suggested that this treatment had little effect on pre-PEA hemodynamics and no effect on the outcome or hemodynamics after PEA [114]. A case series study by Balasubramanian et al. examined 10 patients with symptomatic CTEPH who started treatment with riociguat while waiting to evaluate the operability of PEA or PEA surgery. The duration of riociguat treatment ranged from 2 to 6 months in seven patients, with one patient receiving riociguat for 16 months, with improvement in WHO FC, BNP, 6 MWD, and dyspnea changes [115].

Most studies investigating the use of PAH-targeted treatment of PAH intended for managing patients with CTEPH have shown beneficial effects. However, these results are primarily from uncontrolled observational trials and should be interpreted cautiously. From February 2007 to January 2009, 679 newly diagnosed CTEPH cases were enrolled in the international registry; at the time of CTEPH diagnosis, 37.7% of patients had started treatment for at least one pulmonary arterial hypertension (28.3% operable, 53.8% inoperable) [14]. PEA is the first-choice treatment for CTEPH. Increased drug use in operable patients may delay the referral of patients to PEA.

## 7. Conclusions

CTEPH is a rare disease characterized by PA stenosis and occlusion caused by organized thrombi that results in PH and right-sided heart failure. As described in this review, CTEPH is difficult to understand because it involves a variety of pathologies, including inflammatory cytokines and coagulation abnormalities. Because CTEPH is associated with a high rate of morbidity and mortality, an appropriate therapeutic approach is crucial. Surgical PEA has been established as a curative treatment for operable CTEPH and is recommended by the international clinical practice guidelines. However, the prognosis is extremely poor if the therapeutic intervention is delayed. Therefore, it is necessary to detect CTEPH at the earliest possible stage and perform a PEA. For early diagnosis, elevated BNP levels in blood tests and right heart stress findings on echocardiography should be detected early, and CTEPH should be confirmed by right heart catheterization, pulmonary arteriography CT, and pulmonary ventilation blood flow scintigraphy. There are two types of treatments: PEA, which is a surgical treatment, and BPA or drug therapy, which is a medical treatment. PEA is a curative procedure for which evidence has been accepted and absolutely indicated. However, PEA is often clinically challenging and requires a high level of surgical skill and volume. On the other hand, two main treatment options are available for inoperable CTEPH: BPA, a treatment to physically release stenosis or obstruction of the PA using a balloon catheter, and drug therapy such as riociguat, an oral soluble guanylate cyclase stimulator. As discussed here, the outcomes of BPA or medical therapy have improved dramatically in recent years, and future evidence is expected.

## Figures and Tables

**Table 1 ijms-24-03979-t001:** CTEPH Prediction Score; a Score of More Than Six Points Denotes a high risk.

Points for Score	Score
Unprovoked pulmonary embolism	+6
Known hypothyroidism	+3
Symptom onset greater than 2 weeks before pulmonary embolism diagnosis	+3
Right ventricular dysfunction on CT or echocardiogram	+2
Known diabetes mellitus	−3
Thrombolytic therapy or embolectomy	−3

A Score greater than six points denotes a “high risk” for CTEPH.

**Table 2 ijms-24-03979-t002:** Indication Criteria for PEA.

• Mean PA pressure ≥ 30 mmHg • Pulmonary vascular resistance ≥ 300 dyne·s·cm^−5^ • NYHA/WHO functional class ≥ III • The central end of the PA lesion must be in an area that can be reached surgically • No serious complications (comorbidities)

**Table 3 ijms-24-03979-t003:** PA occlusion morphology (Jamieson disease classification).

Type of PA Occlusion Morphology	Characteristics
Type 1	The presence of a mural thrombus in the main PA or interlobar artery
Type 2	Organized thrombus or intimal thickening on the central side of the regional artery
Type 3	Intimal thickening or fibrotic tissue is present on the peripheral side of the regional artery
Type 4	Lesions in the small arteries

**Table 4 ijms-24-03979-t004:** PA occlusion morphology (University of California San Diego chronic thromboembolism surgical classification).

Surgical Levels	Location of Chronic Thromboembolism (CTE)
Level 0	No evidence of thromboembolic disease in either lung
Level I	CTE starting in the main pulmonary arteries
(Level IC)	(Complete occlusion of one main PA with CTE)
Level II	CTE starting at the level of lobar arteries or in the main descending pulmonary arteries
Level III	CTE starting at the level of the segmental arteries
Level IV	CTE starting at the level of the subsegmental arteries

**Table 5 ijms-24-03979-t005:** Indications for BPA.

① Difficulty in performing PEA	• Cases in which the lesion is below the regional artery, difficult to reach surgically, or proximal to the regional artery, but PEA is not performed due to complications that would interfere with surgery • Cases of residual or recurrent PH after PEA
② Insufficient response to medical treatment	• NYHA/WHO functional class III or higher (mean PA pressure > 30 mmHg or pulmonary vascular resistance > 300 dyne-s-cm^−5^) despite drug therapy
③Explanation and Consent	• The patient (and family members) wishes to use BPA after having been fully informed of the medical condition and the risk-benefit ratio of BPA
④ Exclusion criteria	• Severe multiorgan failure, especially renal dysfunction

## Data Availability

Not applicable.

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
