# Peer review of "Pathophysiology and Treatment of Chronic Thromboembolic Pulmonary Hypertension"

_ijms, 2023, doi:10.3390/ijms24043979_

Round 1

Reviewer 1 Report

Page 1 line14: in medicine any treatment is medical, change medical therapy to pharmaceutical therapy. or drug therapy

Page 1 line 37-40: please provide reference for the involvement of small-vessel abnormalities in CTEPH causation and progression.

Page 1 line 43: change medical to drug

Page 3 line  line 102-103: I don’t understand possible typo? ‘’… and elevations of circulating micovessels…’ How can part of the vasculature be circulating? Or did you mean micro vesicles as in particles, if so please specify.

Page 3 line 125/26 with respect to the Thr312Ala polymorphism mention the correct gene symbol FGA (134820).

Page4 line232: for the non-specialist reader explain up-to-area thrombi I have no idea!

Page 6 line 257: incorrect sentence?:  Therefore, CETPH will first determine the indication for surgery for PEA after the diagnosis is confirmed. Don’t understand what you mean – how can CETPH itself determine anything?  Faulty interpunction?

Page7 table 3:  location of CTE ( please wright out CTE first time when you indtroduce the abbreviation.

Page 8 line 335: change medical to drug

Page 8 line 356: what is a CTO?

Page 9: table1: it is confusing that your review includes 2 tables 1, I’d figure this one is table 4.; change medical to drug.  Try to create an additional space between Consent and ‘ter’ in item 3.

Page 10 line 409: probable typo PAH should read PEA?

Page 11 line456: probable typo:’’ between placebo and placebo groups’

Page 11 line 487/8: introduce 6-MWD as 6 minutes walking distance

Page 12 line 510: please be consistent in your use of 6 MWD and 6-minute walking distance. throughout the document

Author Response

Thank you for comments us to revise our manuscript entitled, “Pathophysiology and treatment of chronic thromboembolic pulmonary hypertension”. We also appreciate the time and effort you and each of the reviewers have dedicated to providing insightful feedback on ways to strengthen our paper. Thus, it is with great pleasure that we resubmit our article for further consideration. We have incorporated changes that reflect the detailed suggestions you have graciously provided. We also hope that our edits and the responses we provide below satisfactorily address all the issues and concerns you and the reviewers have noted.

1.    Page 1 line14: in medicine any treatment is medical, change medical therapy to pharmaceutical therapy. or drug therapy
Response: Thank you for pointing out. As the reviewer pointed out, we noticed some confusion in the use of the terms "medical therapy" and "drug therapy". in this review. Therefore, as per the reviewer's comments, we revised to use the term "drug therapy" in this review (Page 1, line 14).

2.    Page 1 line 37-40: please provide reference for the involvement of small-vessel abnormalities in CTEPH causation and progression.
Response: Thank you for your comments. We will provide references in accordance with your comment (Page 1, line 43).

3.    Page 1 line 43: change medical to drug
Response: Thanks for your comment. I have corrected the same as above (Page 2. line 47).

4.    Page 3 line 102-103: I don’t understand possible typo? ‘’… and elevations of circulating micovessels…’ How can part of the vasculature be circulating? Or did you mean micro vesicles as in particles, if so please specify.
Response: Thank you for your comment. This is a typographical error. It is "platelet microparticles" not "circulating microvessels". I have revised the text significantly to make it clearer.

5.    Page 3 line 125/26 with respect to the Thr312Ala polymorphism mention the correct gene symbol FGA (134820).
Response: I may not have understood the reviewer's comments correctly; I added (rs6050) to Thr312Ala.Thr312Ala (substitution of threonine (Thr) at position 312 with alanine (Ala); rs6050)(Page3, line 135) 

6.    Page5 line232: for the non-specialist reader explain up-to-area thrombi I have no idea!
Response: Thank you for your comment. We have changed "up-to-area thrombi" to " subsegmental or segmental artery thrombosis" per the reviewer's comment (Page 6, line 245).

7.    Page 6 line 257: incorrect sentence?:  Therefore, CETPH will first determine the indication for surgery for PEA after the diagnosis is confirmed. Don’t understand what you mean – how can CETPH itself determine anything?  Faulty interpunction?
Response: As the reviewer pointed out, the interpretation may be poor. Therefore, we have changed “CTEPH will first determine the indication for surgery for PEA after the diagnosis is confirmed” to “after confirming the diagnosis of CTEPH, the indication for surgery for PEA is first determined” (Page 6, line 273).

8.    Page7 table 3:  location of CTE (please wright out CTE first time when you introduce the abbreviation.
Response: Thank you for pointing this out. We have made the change according to the reviewer (Page 8, table 4).

9.    Page 8 line 335: change medical to drug
Response: Thanks for your comment. I have corrected the same as above (Page 10, line 429, 430, 434).

10.    Page 8 line 356: what is a CTO?
Response: Thank you for pointing this out. We have made the change according to the reviewer (Page 9, line 400).

11.    Page 9: table1: it is confusing that your review includes 2 tables 1, I’d figure this one is table 4.; change medical to drug.  Try to create an additional space between Consent and ‘ter’ in item 3.
Response: Thanks for the comment. It was a typographical error. It has been changed to table 5. The figure has been resized for clarity (Page 10).

12.    Page 10 line 409: probable typo PAH should read PEA?
Response: I’m sorry, I could not understand the reviewer's comment. The reference for this text is "A prospective, randomized study of inhaled prostacyclin versus nitric oxide in patients with residual pulmonary hypertension after pulmonary endarterectomy". Therefore, "patients with residual PH after PEA" seems to be correct (Page 11, line 467).

13.    Page 11 line456: probable typo:’’ between placebo and placebo groups’
Response: As the reviewer pointed out, this is a mistake. We have corrected "between placebo and placebo groups" to "between bosentan and placebo groups" (Page 12, line 518).

14.    Page 11 line 487/8: introduce 6-MWD as 6 minutes walking distance
Response: Thanks for your comments. I have standardized the listing from abbreviations to 6-minute walk distance (Page 12, line 530).

15.    Page 12 line 510: please be consistent in your use of 6 MWD and 6-minute walking distance. throughout the document
Response: Thanks for your comment. I have corrected and standardized the same as above (Page 13, line 556).

Again, thank you for giving us the opportunity to strengthen our manuscript with your valuable comments and queries. We have worked hard to incorporate your feedback and hope that these revisions persuade you to accept our submission.

Sincerely,

Naoyuki Otani, M.D. , Ph.D.

Dokkyo Medical University Nikko Medical Center
632 Takatoku, Nikko city, Tochigi, JAPAN
321-2523
TEL: 0288-76-1515

Reviewer 2 Report

The new PH guidelines should be by all means referenced. They were published in 2022.  All values and data (like treatment duration etc ) shoud be corrected according to the new guidelines.

No new medications and clinical trials (like MACITEPH) are discussed.

Author Response

Thank you for comments us to revise our manuscript entitled, “Pathophysiology and treatment of chronic thromboembolic pulmonary hypertension”. We also appreciate the time and effort you and each of the reviewers have dedicated to providing insightful feedback on ways to strengthen our paper. Thus, it is with great pleasure that we resubmit our article for further consideration. We have incorporated changes that reflect the detailed suggestions you have graciously provided. We also hope that our edits and the responses we provide below satisfactorily address all the issues and concerns you and the reviewers have noted.

1.    The new PH guidelines should be by all means referenced. They were published in 2022.  All values and data (like treatment duration etc ) shoud be corrected according to the new guidelines.
Response: We perused the new PH guidelines and revised the manuscript, as noted by the reviewers. We referred to Chapter 10 CTEPH (group 4) and Chapter 14 in the 2022 ESC/ERS Guidelines.

2.    No new medications and clinical trials (like MACITEPH) are discussed.
Response: Thank you for your advice. We have made the addition with reference to the clinical trial such as Macitentan (6.2.2. Macitentan) you mentioned.

Again, thank you for giving us the opportunity to strengthen our manuscript with your valuable comments and queries. We have worked hard to incorporate your feedback and hope that these revisions persuade you to accept our submission.

Sincerely,

Naoyuki Otani, M.D. , Ph.D.

Dokkyo Medical University Nikko Medical Center
632 Takatoku, Nikko city, Tochigi, JAPAN
321-2523
TEL: 0288-76-1515

Reviewer 3 Report

I should congratulate authors for a well written article. However I will like to include some more additional points for completion.

1) Management of acutely symptomatic patients with CTPH.

2) Role of mechanical circulatory devices in CTPH.

3) Role of Heart lung transplantation in selected patients.

4) Recent guidelines for management of CTPH- Both medical & surgical options.

I am sure addition of them will make the article complete.

Author Response

Thank you for comments us to revise our manuscript entitled, “Pathophysiology and treatment of chronic thromboembolic pulmonary hypertension”. We also appreciate the time and effort you and each of the reviewers have dedicated to providing insightful feedback on ways to strengthen our paper. Thus, it is with great pleasure that we resubmit our article for further consideration. We have incorporated changes that reflect the detailed suggestions you have graciously provided. We also hope that our edits and the responses we provide below satisfactorily address all the issues and concerns you and the reviewers have noted.

1) Management of acutely symptomatic patients with CTPH.
Response: Thank you for your comment. We have added an addendum on CTEPH in the context of acute PE with reference to “Delcroix M et al. ERS statement on chronic thromboembolic pulmonary hypertension".

2) Role of mechanical circulatory devices in CTPH.
Response: Thank you for your advice. I have added a note about the role of mechanical circulatory devices in CTEPH, especially ECMO (Page 7).

3) Role of Heart lung transplantation in selected patients.
Response: Thank you for your advice. I have added a note about the roles of lung transplantation for CTEPH (Page 7).

4) Recent guidelines for management of CTPH- Both medical & surgical options.
Response: Thank you for your advice. We perused the new PH guidelines and revised the manuscript, as noted by the reviewers (Page 7, line 335 etc).

Again, thank you for giving us the opportunity to strengthen our manuscript with your valuable comments and queries. We have worked hard to incorporate your feedback and hope that these revisions persuade you to accept our submission.

Sincerely,

Naoyuki Otani, M.D. , Ph.D.

Dokkyo Medical University Nikko Medical Center
632 Takatoku, Nikko city, Tochigi, JAPAN
321-2523
TEL: 0288-76-1515

Round 2

Reviewer 2 Report

The new ERS/ESC guidelines clearly stick to the new numbers for PH diagnosis (mean PAP>20 mmHg). It should be mentioned in the diagnosis section.

Author Response

Thank you for your comment. I think it is an important comment. We greatly appreciate your peer review. We have revised the manuscript according to the reviewers' comments. We would appreciate it if you could check it (page 1, line 22-26 and page 5, line 210).
